# The Role of Interventional Radiology in the Diagnosis and Treatment of Pulmonary Arteriovenous Malformations

**DOI:** 10.3390/jcm11216282

**Published:** 2022-10-25

**Authors:** Chee Woei Yap, Bernard B. K. Wee, Sze Ying Yee, Vincent Tiong, Yi Xiu Chua, Lycia Teo, Rahul Lohan, Amos Tan, Pavel Singh, Prapul Chander Rajendran, Cunli Yang, Yong Chen Yee, Gopinathan Anil, Shao Jin Ong

**Affiliations:** 1National University Hospital, National University Health Systems, Singapore 119228, Singapore; 2Ng Teng Fong General Hospital, National University Health Systems, Singapore 609606, Singapore

**Keywords:** interventional radiology, endovascular surgery, embolization, percutaneous interventions, image-guided interventions, pulmonary arteriovenous malformations, PAVM, hereditary hemorrhagic telangiectasia, right to left shunt

## Abstract

Pulmonary arteriovenous malformations (PAVMs) are uncommon, predominantly congenital direct fistulous connections between the pulmonary arteries and pulmonary veins, resulting in a right to left shunt. Patients with PAVMs are usually asymptomatic with lesions detected incidentally when radiological imaging is performed for other indications. In this review, we discuss the classification and radiological features of PAVMs as well as their treatment and follow-up options, with a particular focus on percutaneous endovascular techniques and the evolution of the available equipment for treatment.

## 1. Introduction

Pulmonary arteriovenous malformations (PAVMs) are uncommon and predominantly congenital in nature. They are direct fistulous connections between pulmonary arteries and pulmonary veins. PAVMs bypass the normal pulmonary capillary bed resulting in a right-to-left shunt [1]. A nidus is usually present. Although the majority of patients with PAVMs are asymptomatic, undiagnosed patients can present later in life with serious complications such as paradoxical embolism, stroke, myocardial infarction, cerebral abscesses, and massive hemoptysis [2].

PAVMs affect as many as 1 in 2600 individuals [3,4], with a female-to-male ratio of 1.67:1 [5]. Approximately 80% of PAVMs are associated with hereditary hemorrhagic telangiectasia (HHT). Conversely, 50% of patients with HHT have PAVMs, which are often multiple in nature [6]. HHT is an autosomal dominant disorder with an estimated incidence of 20 per 100,000 population [2]. HHT is characterized by a classic triad of cutaneous telangiectasia, recurrent epistaxis, and family history of HHT. Patients with HHT are at risk of venous thrombosis. Most of them tolerate anticoagulant and antiplatelet therapy well despite the hemorrhagic nature of the disease.

Acquired causes of PAVMs include prior surgery, trauma, infection, hepatopulmonary syndrome, congenital heart disease (post-Glenn or modified Fontan procedures) [7], and metastatic disease [8].

PAVMs can be classified into three types [9]:(1)Simple type: commonest, up to 80%, with a single segmental artery feeding the malformation;(2)Complex type: with multiple segmental feeding arteries;(3)Diffuse type: rare, with hundreds of malformations [10].

Some patients can present with a combination of simple and complex PAVMs.

Most patients with PAVMs remain asymptomatic. Those who suffer from sequelae of right-to-left shunting (e.g., shortness of breath on exertion, cyanosis, and fatigability) or paradoxical embolisms (e.g., stroke and cerebral abscess) require intervention and treatment.

## 2. Radiological Diagnosis

Most PAVMs, being asymptomatic, are detected on radiological investigations done for other indications.

Regarding chest radiographs, typical PAVMs appear as well-defined soft tissue lesions with vessels directed towards the hilum (Figure 1A). However, smaller lesions may be undetectable due to the low sensitivity of the study.

Contrast-enhanced computed tomography (CT) is the modality of choice for the characterization of PAVMs. Careful tracing, especially on fine cuts can show one or more feeding pulmonary arteries leading to a nidus, as well as one or more draining pulmonary veins leading away from the nidus (Figure 1B). There is a predilection for their location in the middle and lower lobes, as well as the lingula [11]. The ground–glass opacities between the artery and vein are due to volume averaging of the microscopic telangiectasia. Together with different clinic presentations, these unique imaging features serve to differentiate PAVMs from other lung lesions, such as pulmonary hemorrhage, fibrosing mediastinitis, venovenous or arterial collaterals, hepatopulmonary vessels, Sheehan vessels, and pulmonary vein varix.

Catheter pulmonary angiogram is the gold standard diagnostic modality (Figure 2B). It is prudent to obtain a prior CT pulmonary angiogram (Figure 2A) in order to vessel map for the purposes of treatment planning.

The second International Guidelines for the Diagnosis and Management of Hereditary Hemorrhagic Telangiectasia, suggests the use of transthoracic contrast echocardiography as the initial screening test for PAVMs [12]. Transcranial doppler (TCD) is an alternative method to screen for PAVMs. The transthoracic contrast echocardiography and TCD do not require ionizing radiation. Both involve intravenous injections of microbubbles (agitated saline mixture of patient’s blood, saline and air) as a sonographic contrast agent. Microbubbles should not reach the left atrium under normal circumstances as they will be filtered out by the pulmonary capillary bed. Detection of microbubbles in the left heart in both methods suggests the presence of right-to-left shunts. For transthoracic contrast echocardiography, intracardiac or extracardiac shunt differentiations are based on the timing of when the bubbles first appear in the left atrium. If a right-to-left shunt is present within three to six cardiac cycles after agitated saline injection, this is likely due to intracardiac shunts whilst bubbles detected in the left atrium after six cardiac cycles suggest intrapulmonary shunts [13]. Detection of microbubbles by TCD can be quantified according to Spencer’s grading. It has been found that patients with Spencer grade < 3 on TCD will also screen negative for PAVMs on CT thorax (100% negative predictive value). Patients with positive findings on TCD may have a negative CT which may indicate the presence of multiple microscopic telangiectasias. It was also found that TCD has 100% sensitivity to detect right to left shunts but a relatively low specificity of 58% [14].

Screening of intracranial and hepatic AVMs in adults with definite or suspected HHT is recommended. Ultrasound doppler, multiphasic contrasted CT, or an MRI with contrast can be performed for screening of hepatic AVMs whilst MRI brain with or without contrast administration can be utilized to screen for cerebral AVMs [12].

## 3. Management of PAVM

The British Thoracic Society recommends that all patients with radiographically visible PAVMs be considered for referral to interventional radiology for embolization and follow-up [3]. Embolization is recommended for asymptomatic patients in view of the complications related to right-to-left shunts, such as paradoxical emboli, as well as to improve oxygenation and effort tolerance. The Cardiovascular and Interventional Radiological Society of Europe (CIRSE) recommends treatment for PAVMs if any one of three criteria is met [15]:(1)Any PAVM (solitary or multiple) with a feeding artery that is 2 mm or larger;(2)Measurable increase in the size of the PAVM;(3)Paradoxical emboli or symptomatic hypoxemia.

Embolization is the preferred treatment given its high success rate [12]. When comparing surgical intervention to percutaneous transcatheter embolization for PAVMs, Nagano et al. reported that patients who underwent surgery had a statistically significantly higher proportion of composite complications (6.9% vs. 2%, *p* = 0.027) and required a longer hospital stay (6 days vs. 2 days; *p* < 0.01). On the other hand, the surgical group resulted in a significantly lower rate of reintervention (2.1% vs. 8.3% at 2 years; *p* < 0.01) [16].

Surgical treatment is generally reserved for cases involving [16]:(1)Diffuse lobar or segmental PAVMs;(2)Complex PAVMs not amenable to embolization;(3)PAVMs with large feeding arteries not amenable to embolization;(4)Practices or regions that lack an interventional radiology service;(5)Patients with contraindication to iodinated contrast medium.

Endovascular embolization of PAVMs involves the cannulation of a peripheral vein (usually the common femoral vein) followed by selective cannulation of the pulmonary artery and super-selective cannulation of the feeding arteries to the PAVMs. Ideally, the nidus followed by all the feeding arteries should be embolized.

Previously, Amplatzer™ Vascular Plug (AVP; Abbott Cardiovascular, Abbott Park, Chicago, IL, USA) was commonly used (Figure 2C) in our practice, as it allowed for controlled deployment with the possibility for readjustments prior to plug release. The smallest currently available Amplatzer™ vascular plug is 3 mm, which needs to be introduced through a sheath or catheter with at least a 0.38-inch lumen diameter.

The more recently available Medtronic (Dublin, Ireland) microvascular plug (Medtronic MVP-3Q) supports the embolization of arteries as small as 1.5–3.0 mm and can be deployed via microcatheters.

While vascular plugs and microvascular plugs allow for rapid occlusion of PAVMs, they may not be suitable for completely occluding the nidus due to vascular anatomy and the shape limitations of the plugs. This results in an increased risk of delayed reperfusion of the nidus despite initial successful angiographic appearances immediately post-embolization.

One solution would be to use a combination of vascular plugs and embolization coils to pack the nidus. This has been shown to reduce reperfusion rates as compared to vascular plug use alone [5].

Pushable embolization coils have been long established for use in vessel embolization. While generally safe for use in standard vascular embolization, they do not allow for the option of repositioning once the coil has been introduced into the target catheter. Undersized coils may migrate past the intended site of embolization into a more distal aspect of the vessel target. In the context of an arterial to venous shunt, the effect of unintended coil migration is significantly amplified with the possibility of an undersized coil migrating directly into the pulmonary venous system and subsequently into the left heart circulation with catastrophic consequences.

Detachable coils are a group of newer-generation devices that allow for controlled deployment with the option of coil retrieval and repositioning. These have significantly reduced the risk of non-target embolization. The packing of the coils can be done in a more controlled manner and the delivery wire can be detached only when the packing is satisfactory. Different detachment mechanisms are available:(1)Electrical detachments, such as AZUR HydroCoil Detachable (Terumo Interventional Systems, Somerset, NJ, USA),(2)Mechanical latch and hook mechanisms, such as Interlock and IDC detachable embolization coils (Boston Scientific, Marlborough, MA, USA)(3)Ball and socket mechanical detachment mechanisms, such as Ruby Coils (Penumbra, Alameda, CA, USA) and Concerto coils (Medtronic, Santa Rosa, CA, USA).

There are many options for long sheaths to access the pulmonary arteries from a femoral puncture. Since 2015, our institutions have used intracranial access catheters. Using an intracranial guiding catheter offers the best of both worlds with good distal flexibility and adequate mid and proximal support. It is more compliant to navigate through the right heart into the pulmonary artery without straightening and straining the vasculature. This property can potentially reduce the significant risks of cardiac arrhythmia and vascular injury. In addition, neurovascular access catheters, such as the Benchmark™ (Penumbra, Alameda, CA, USA), are of sufficient inner diameter (6Fr) and can be safely advanced into the segmental and sometimes even sub-segmental branches, thereby providing stable access for embolization.

Concurrently, we have also moved from using a combination of a vascular plug with pushable embolization coils to using long lengths of soft detachable coils (Figure 3). These newer-generation coils allow for much more controlled and tighter packing of the proximal draining veins, the nidus, and the feeding arteries, thereby reducing the risk of reperfusion reoccurrence. An additional benefit of using the softer detachable coils is that we are now able to access the smaller, more peripheral (Figure 3C) or tortuous PAVMs, which were previously not accessible by the traditional vascular plugs due to the rigidity of both the plug and their deployment system.

With an incidence of 22%, reperfusion was previously the most common problem post-endovascular treatment of PAVMs. Reperfusion was most commonly a result of the recanalization of the embolized artery. The second most common cause was the development of accessory vessels (up to 38%) [5]. Milic et.al. demonstrated that reperfusion was associated with the use of a single coil, oversized coils, coil placement more than 1 cm from the aneurysm, and increased feeding artery size [17]. The use of newer-generation softer and longer detachable coils is likely to mitigate this.

## 4. Complications of Endovascular Treatment

Distal migration of coils and non-target embolization are known complications of pushable coils. These can be avoided with the usage of detachable coils which offer an improved safety profile. However, in the absence of appropriately sized detachable coils, several techniques such as the anchor technique, scaffold technique or balloon-occlusion technique can be applied to increase pushable coil stability and thus prevent distal migration of coils [15]. The anchor technique involves advancing at least 2 cm of the coil into a side branch (which is normally sacrificed) and the rest of the coil is then deployed proximally followed by additional coil packing. In the scaffold technique, a high radial force coil (scaffold) is placed initially, then soft coils may be packed within the scaffold. A balloon catheter may be useful to decrease the flow through the PAVMs, promoting and allowing thrombus formation around the coils, hence reducing the risk of distal migration. The balloon catheter is inflated proximal to the target feeding artery or nidus before deployment of the pushable coils and removed once the coils have attained stability in their intended position.

In view of the right to left shunt, another potentially life-threatening complication is systemic air embolism [18]. Air embolisms can result from air bubbles trapped in the catheter hub being advanced along with wire or aspirated from the end of a catheter open to the air. Air embolisms can be prevented with wire and catheter exchanges performed using a “water bath technique”. This involves the use of a Tuohy Borst Y Connector attached to a pressurized saline bag at 300 mmHg via an intravenous infusion set. The pressurized saline is attached to the side arm of the connector and the rate of flow can be controlled via the V-track controller of the infusion set. Permanent positive pressure flow via the side arm together with the Touhy Borst valve will flush any air bubbles through the back end of any catheter/microcatheter/wire inserted through it and prevent any bubbles from being pushed into the system.

Post-procedure, the most common early complications are pleuritic chest pain [18] and blood-stained sputum. These are usually self-limiting [5].

While there have been concerns for increased pulmonary arterial pressure following embolization of PAVMs, a reasonably large study of treated PAVMs by Shovlin et al. reported pulmonary arterial pressure was not increased significantly by PAVM embolization [19].

## 5. Post-Embolization Follow-Up

Follow-up cross-sectional imaging together with a clinical review is recommended to be performed at approximately three to six months post-embolization to assess clinical, physiological, and radiological response (Figure 4). A commonly used definition for assessing recanalization is a less than 70% reduction in the diameter of the nidus/venous sac and draining vein on contrast-enhanced CT. However, this is of limited utility as it has a low specificity for predicting recanalization (draining vein, 7.3%; venous sac/nidus, 41.7%) [20]. Additionally, with some of the newer and softer detachable coils, proceduralists aim for complete embolization of the nidus to reduce recurrence. This leads to significant bean hardening streak artifacts from the coils in the nidus on CT and limits accurate evaluation of nidus and draining vessel size and flow. While the major CT brands such as GE, Siemens and Philips do supply metal artifact reduction reconstruction algorithms as an additional software option, the proximity between tightly packed embolic coils and the draining veins still poses a significant limitation to the accurate measurement of vessel diameter changes.

As an alternative to CT assessment, both contrast-enhanced and non-contrast-enhanced magnetic resonance angiography (MRA) have been described for the assessment of PAVMs. Contrast-enhanced single-shot whole thoracic time-resolved MR angiography (TR-MRA) has been reported to be helpful in the assessment of the recanalization of the embolized PAVMs [21]. TR-MRA demonstrated high diagnostic accuracy (sensitivity and specificity of >90%) and it is relatively insusceptible to metal artifacts. It has the additional advantage of providing the hemodynamic characteristics of untreated or re-perfused PAVMs. The sets of dynamic contrast-enhanced images acquired in TR-MRA are similar to the dynamic series in conventional angiography images. Non-contrast-enhanced MRA using Time-SLIP (time-spatial labeling inversion pulse) technique has been described by Hamamoto et.al. with promising sensitivity and specificity of 100% for detection of PAVMs reperfusion [22]. While MRA is hugely promising as an accurate and radiation-free modality for the assessment and follow-up imaging of PAVMs, resource limitations arising from both long acquisition timing and costs have limited the adoption of MRA as the primary imaging modality for many centers.

## 6. Conclusions

Endovascular embolization should be considered a first-line treatment method for PAVMs due to its lower procedure-related complications and shorter hospital stay compared to surgical treatment. Although there are concerns about an increased risk of delayed reperfusion of the PAVM in patients treated by embolization, the introduction and adaptation of newer-generation access and embolization devices are expected to reduce reperfusion rates. In addition, the technical success rate of endovascular treatment is expected to improve even with more peripheral and smaller PAVMS.

## Figures and Tables

**Figure 1 jcm-11-06282-f001:**
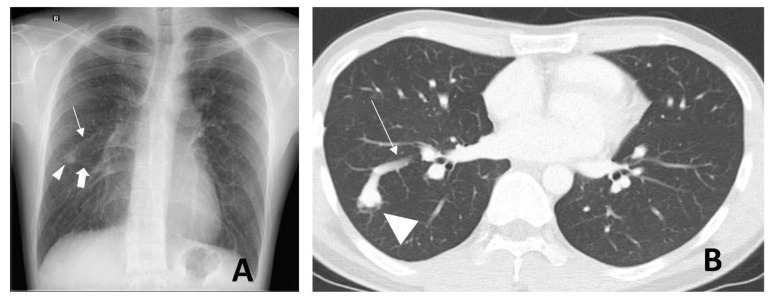
A 26-year-old male with a pulmonary arteriovenous malformation (PAVM) at the apical segment of the right lower lobe. Posterior–anterior chest radiograph (**A**) shows a right mid-zone pulmonary nodular opacity (arrowhead) with two cord-like structures arising from the medial margin of the lesion and directed toward the hilum. These are the feeding artery (thin arrow) and the draining vein (thick arrow). Axial contrast-enhanced computed tomography (**B**); lung window shows the same right lower lobe PAVM (arrowhead) with the partially-imaged feeding artery (thin arrow).

**Figure 2 jcm-11-06282-f002:**
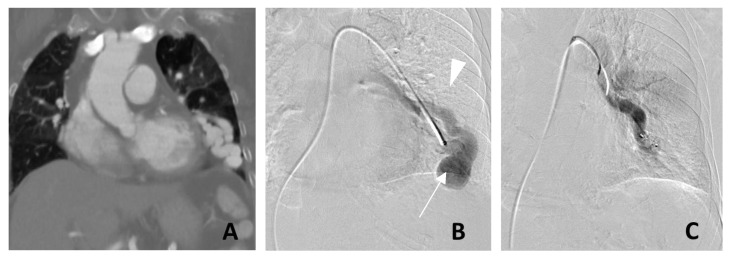
Left lower lobe pulmonary arteriovenous malformation (PAVM) treated with a single Amplatzer™ vascular plug. Coronal contrast-enhanced computed tomography (**A**) demonstrates a large arteriovenous malformation in the left lower lobe with associated cardiomegaly. Digital subtraction angiogram (**B**) demonstrated a high flow PAVM with a large draining vein. (**C**) Complete angiographic cessation of flow following successful embolization with a Type II Amplatzer ™ Vascular Plug. The nidus (arrow) and the draining vein (arrowhead) are no longer opacified on the post-embolization digital subtraction angiogram.

**Figure 3 jcm-11-06282-f003:**
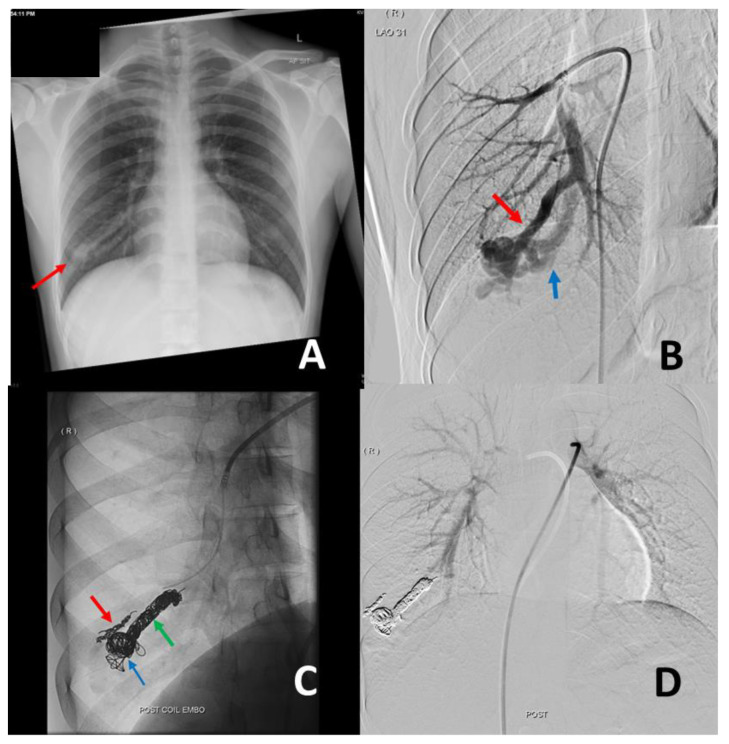
A 28-year-old male manual laborer presented with increasing shortness of breath and saturations of 90% on room air. Antero-posterior chest radiograph (**A**) demonstrated an ill-defined nodule (red arrow) in the right lower zone. Digital subtraction angiogram (**B**) confirmed the presence of a pulmonary arteriovenous malformation (PAVM) in the right lower lobe with a dominant feeding artery (red arrow) and a single draining vein (blue arrow). This was embolized (**C**) with a combination of Ruby™ coils (three 10 mm × 35 cm coils, one 8 mm × 60 cm coil, and one 60 cm packing coil) and an Interlock™ coil (6 mm × 20 cm) in the nidus (blue arrow) and the dominant feeding artery (green arrow). An accessory feeding branch (red arrow) was embolized with a Ruby™ coil (3 × 20 mm). Post-procedure angiogram (**D**) via the main pulmonary artery confirms complete occlusion of the PAVM.

**Figure 4 jcm-11-06282-f004:**
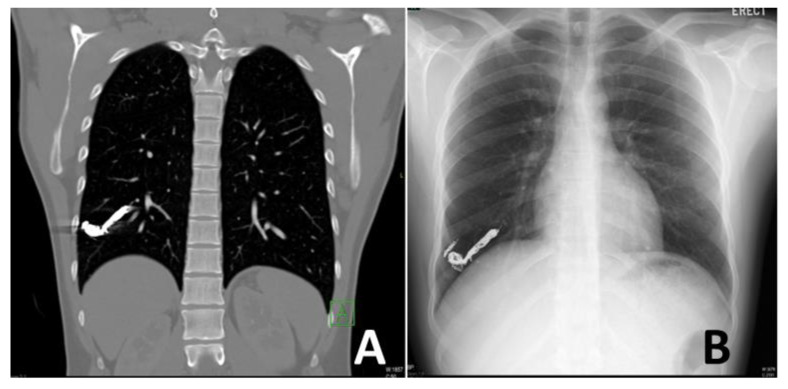
Three-month follow-up imaging of the patient treated in Figure 3. The patient was well with no residual shortness of breath. His saturation on room air also increased to 98%. The coronal contrast-enhanced computed tomography (**A**) and posterior-anterior chest radiograph (**B**) demonstrate the tightly packed coils with no evidence of flow recurrence or any new complications.

## Data Availability

Not applicable.

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
