# Peer review of "The Role of Interventional Radiology in the Diagnosis and Treatment of Pulmonary Arteriovenous Malformations"

_jcm, 2022, doi:10.3390/jcm11216282_

Round 1

Reviewer 1 Report

I would like to thank the authors for the opportunity to review their work, the paper entitled "The role of interventional radiology in the diagnosis and treatment of Pulmonary Arterio-Venous Malformations".

As I understand the paper it represents a case-based literature review, therefore, it should be identified in the title as such.

The cases are exceptionally interesting; however their description should include in the case of CT images (if it was the case): the usage of contrast agents (CT angiography if this was the case), the slice orientation (axial/coronal/sagital/oblique).

Figure 3A and 4(A and B) should remove any identifiable items from the image

Author Response

Reviewer Comment: As I understand the paper it represents a case-based literature review, therefore, it should be identified in the title as such.

Authors reply: Thank you. This is a good point but the authors feel that this will add too much length to an already long title. We hope that the review would be able to accept the current title.

Reviewer Comment: The cases are exceptionally interesting; however their description should include in the case of CT images (if it was the case): the usage of contrast agents (CT angiography if this was the case), the slice orientation (axial/coronal/sagital/oblique).

Authors Reply: Thank you for the advice. The changes have been made as per your advice. 

Reviewer comment: Figure 3A and 4(A and B) should remove any identifiable items from the image

Authors reply: These were initially ignored as they only stated a system generated "dummy name". However we have followed the reviewer's advice and cropped or covered any potential wording suggestive of identification, real of system generated. 

Reviewer 2 Report

General comments:

Overall, this review article is well organized. However, considering that this manuscript was submitted as a review article for an international journal, I feel it is necessary to mention the international guidelines. In addition, I think some revisions are needed.

Specific comments:

1. Introduction

(p2, L55) PAVMs can be classified into three types:

-Please cite the original literature of the PAVM classification (AJR Am J Roentgenol. 1983 Apr;140(4):681-6. doi: 10.2214/ajr.140.4.681.)

2. Radiological Diagnosis

(p3, L96-103) Transcranial doppler (TCD) is... relatively low specificity of 58% [11].

-In general, transthoracic contrast echocardiography has been recommended for screening for PAVM rather than transcranial Doppler. Please rewrite this section with reference to the International Guidelines for the Diagnosis and Management of Hereditary Hemorrhagic Telangiectasia (Ann Intern Med. 2020 Dec 15;173(12):989-1001).

(p2, L82) Figure 1.  A 26-year-old gentleman

-I think it’s “A 26-year-old male”

3. Management of PAVM:

(p3, L107-109) British Thoracic Society recommends that... follow up [3].

-I think that the authors should also mention the international PAVM guidelines, such as HHT guidelines (Ann Intern Med. 2020 Dec 15;173(12):989-1001) and CIRSE guidelines (Cardiovasc Intervent Radiol. 2020 Mar;43(3):353-361).

(p5, L182-185) With an incidence of 22% ...development of accessory vessels (up to 38%).

-Please append the reference(s).

4. Complications of endovascular treatment:

(p6, L201-203) Distal migration of coils ...improved safety profile.

- The use of a balloon catheter may be useful to prevent migration of coils and thrombus during embolization. If you are using this method, why not describe it?

(p6, L206) “water bath technique”

-Please append a brief description.

5. Post Embolization follow up:

-Please briefly describe the evaluation method of reperfusion on CT. Evaluation using CT may be difficult due to metallic artifacts originated from the coils, particularly in the case with nidus embolization. Recently, the utility of contrast or non-contrast magnetic resonance angiography with less metal artifacts have been reported in the evaluation of reperfusion of embolized PAVMs. Please discuss these methods.

Author Response

Specific comments:

  1. Introduction

(p2, L55) PAVMs can be classified into three types:

-Please cite the original literature of the PAVM classification (AJR Am J Roentgenol. 1983 Apr;140(4):681-6. doi: 10.2214/ajr.140.4.681.)

 This has been done. Thank you.

  1. Radiological Diagnosis

(p3, L96-103) Transcranial doppler (TCD) is... relatively low specificity of 58% [11].

-In general, transthoracic contrast echocardiography has been recommended for screening for PAVM rather than transcranial Doppler. Please rewrite this section with reference to the International Guidelines for the Diagnosis and Management of Hereditary Hemorrhagic Telangiectasia (Ann Intern Med. 2020 Dec 15;173(12):989-1001).

Thank you. This section has been changed and both Trans Thoracic echocardiography and TCD has been described in more detail.

(p2, L82) Figure 1.  A 26-year-old gentleman

-I think it’s “A 26-year-old male”

Agreed with reviewer. Apologies for oversight. This has been changed.

  1. Management of PAVM:

(p3, L107-109) British Thoracic Society recommends that... follow up [3].

-I think that the authors should also mention the international PAVM guidelines, such as HHT guidelines (Ann Intern Med. 2020 Dec 15;173(12):989-1001) and CIRSE guidelines (Cardiovasc Intervent Radiol. 2020 Mar;43(3):353-361).

 Thank you for the insight. This section has been updated.

(p5, L182-185) With an incidence of 22% ...development of accessory vessels (up to 38%).

-Please append the reference(s).

 Apologies for oversight. This has now been added.

  1. Complications of endovascular treatment:

(p6, L201-203) Distal migration of coils ...improved safety profile.

- The use of a balloon catheter may be useful to prevent migration of coils and thrombus during embolization. If you are using this method, why not describe it?

 Thank you for suggestion. This has been added

(p6, L206) “water bath technique”

-Please append a brief description.

  Thank you for suggestion. How we do this has been added

  1. Post Embolization follow up:

-Please briefly describe the evaluation method of reperfusion on CT. Evaluation using CT may be difficult due to metallic artifacts originated from the coils, particularly in the case with nidus embolization. Recently, the utility of contrast or non-contrast magnetic resonance angiography with less metal artifacts have been reported in the evaluation of reperfusion of embolized PAVMs. Please discuss these methods.

Thank you for advice. We have now updated the CT evaluation section and appended a MRA evaluation section for this section.